# Resistance and Pathogenicity of *Salmonella* Thompson Isolated from Incubation End of a Poultry Farm

**DOI:** 10.3390/vetsci9070349

**Published:** 2022-07-11

**Authors:** Jingju Zhang, Jing Liu, Chen Chen, Yufeng Wang, Xiaojie Chen, Xiubo Li, Fei Xu

**Affiliations:** National Feed Drug Reference Laboratories, Institute of Feed Research, Chinese Academy of Agricultural Sciences, Beijing 100081, China; daju_18@163.com (J.Z.); liujing0415-@outlook.com (J.L.); chen_vet@foxmail.com (C.C.); xingqiti@163.com (Y.W.); xjchen2009@163.com (X.C.); lixiubo@caas.cn (X.L.)

**Keywords:** *Salmonella* Thompson, cgMLST, *mcr-9*, antibiotic resistance, chicks

## Abstract

**Simple Summary:**

Non-typhoid *Salmonella* is the general term of *Salmonella* other than typhoid and paratyphoid, which often causes foodborne gastroenteritis in humans, but some serotypes have been proved to be pathogenic to poultry. *Salmonella* Enterica and *Salmonella* Typhimurium are the common serotypes pathogenic to poultry and have been systematically studied, but other serotypes have rarely been studied. During *Salmonella* surveillance in farms, we discovered by chance that *Salmonella* Thompson, a common non-typhoid *Salmonella*, is also pathogenic to avian embryos. Therefore, this study aimed to explore antimicrobial resistance and pathogenicity of clinical S. Thompson. Firstly, we found that the core-genome multilocus sequence typing of 14 clinical *S.* Thompson was consistent with two strains of S. Thompson from humans in China. Secondly, the antimicrobial resistance gene analysis demonstrated that all strains carried the polymyxin resistance gene *mcr-9*, which had not appeared resistance phenotype. Meanwhile, many essential virulence genes were also found in each *S.* Thompson isolate. Finally, the bacterial inoculation experiment revealed that clinical *S.* Thompson was highly pathogenic to newborn chicks after yolk sac inoculation. This study suggests that *Salmonella* Thompson can circulate between humans and poultry farms and transmit drug resistance genes and demonstrated that *Salmonella* Thompson is highly pathogenic to chicks and should be guarded against in the hatching stage of poultry farms.

**Abstract:**

*Salmonella* Thompson, an important foodborne pathogen, is rarely found to be pathogenic to poultry. Accidentally, *S.* Thompson was found to be pathogenic to embryos of white feather broiler at a poultry farm in China. Therefore, this study aimed to explore antimicrobial resistance and pathogenicity of clinical *S.* Thompson isolated from dead poultry embryos. The phylogenetic tree based on 16S rRNA and seven housekeeping genes showed that the 14 clinical *S.* Thompson were closely related. The core-genome multilocus sequence typing of 14 clinical *S.* Thompson based on whole-genome sequencing was cgST-12774, consistent with the only two strains of *S*. Thompson from humans in China as reported in the NCBI database. The antimicrobial resistance gene analysis demonstrated that all strains carried *aac(6′)-Iaa* and the polymyxin resistance gene *mcr-9*. Antimicrobial sensitivity tests for 18 antibiotics showed that *S.* Thompson isolates displayed resistance against streptomycin (100%), ampicillin (35.7%), and doxycycline (14.3%), but sensitivity to polymyxin B, proving that the *mcr-9* gene had not appeared resistance phenotype. Virulence genes *Salmonella* pathogenicity island (SPI) SPI1-5, type I fimbriae gene (*fimA*), flagellar assembly genes (*bcfC*, *flhD*, *fliA*, *fliC*, *fljB*, *flgK*, and *lpfC*), and other virulence genes (*iroN*, *pagC*, and *cigR*) were found in each *S.* Thompson isolate. Additionally, the bacterial inoculation experiment with 1-day-old chicks revealed that clinical *S.* Thompson was highly pathogenic to newborn chicks after yolk sac inoculation. This study highlighted that the *S.* Thompson isolated from poultry embryos and the *S.* Thompson causing human foodborne diarrhea in some parts of China belong to the same cgMLST typology (cgST-12774) and showed the pathogenicity of this clinical *S.* Thompson to chicks.

## 1. Introduction

Non-typhoid *Salmonella* (NTS) is a zoonotic foodborne pathogen with flagella that causes acute gastroenteritis in humans and systemic infections in poultry [1,2], causing considerable economic losses to poultry farms [3,4]. Poultry products in the food production supply chain are frequently associated with human salmonellosis cases and are an important cause of NTS transmission between poultry farms and humans [5]. Additionally, NTS is an important repository of antimicrobial resistance genes, posing a great challenge to public health and security [6,7].

With the development of sequencing technology, the means of surveillance for Salmonella has rapidly improved. In recent years, whole-genome sequencing (WGS) has gradually replaced traditional molecular methods such as pulsed-field gel electrophoresis (PFGE) as a new way to track the spread of bacterial infectious diseases due to its ability to obtain complete and detailed genetic information of a species [8,9,10]. Both antimicrobial resistance and virulence factors can be studied using WGS. WGS-based core-genome multilocus sequence typing (cgMLST) constructs high-resolution species typing with high-density core genomes as markers, which compensates for the inability of traditional MLST schemes to distinguish between high recombination levels, low diversity, and differences in monoclonal strains [11].

In China, *Salmonella* Enteritidis and *Salmonella* Typhimurium are the most predominant serovars of NTS in poultry production and human clinical isolates, and *Salmonella* Derby, *Salmonella* Rissen, *Salmonella* Infantis, *Salmonella* Thompson, and *Salmonella* Newport are also frequently isolated serovars of NTS [12]. However, most studies on avian-derived NTS have focused on the investigation of antimicrobial resistance, and few studies have been conducted on the avian pathogenicity of NTS other than on *Salmonella* Enteritidis and *Salmonella* Typhimurium [13]. Therefore, few reports exist to assess the potential risk of different serotypes of NTS on poultry farms. *S.* Thompson, which causes significant white feather broiler embryonic mortality, was detected at hatching at a broiler farm while our laboratory was monitoring *Salmonella* at each end of the broiler production chain, which covered all stages from farm to slaughter, processing, and distribution. Considerable literature documents that *S.* Thompson has caused several international outbreaks of foodborne diseases [14,15]. The most recent was a large-scale gastroenteritis outbreak reported in South Korea in September 2018, with 2207 people infected across 10 schools [16]. *S.* Thompson has been monitored in poultry farms in many regions of China in recent years [1], with a significant regional increase in its prevalence [17,18]. *S.* Thompson also has been reported to exhibit strong environmental resistance and is able to survive on fresh vegetables, meat, various processed foods, and even pickled foods, with the highest isolation rate in chicken [19]. At the same time, *S.* Thompson, like other *Enterobacteriaceae*, has the ability to continuously acquire and spread antimicrobial resistance genes [20], which may limit the clinical use of both human and veterinary drugs.

*S.* Thompson is more common in human diarrhea diseases, but the poultry diseases caused by it are rarely reported, and systematic studies are even rarer. Therefore, this study aims to explore the relationship between *S.* Thompson from poultry and human and to deeply study the resistance and virulence factors of *S.* Thompson to poultry (Figure 1).

## 2. Materials and Methods

The technical route of this study is shown in Figure 1.

### 2.1. Strains

During the monitoring of *Salmonella*, 384 suspected *Salmonella* isolates were collected from Shandong Province and Beijing, China, covering stages of incubation, breeding, processing, and circulation. Samples collected included chicken embryos, chicks, and raw chicken meat. During the sampling, 14 *Salmonella* spp. for this study of these isolates were isolated from an outbreak of dead chicken embryos in the hatchery of a poultry farm in Dezhou, Shandong Province. The isolation and identification of *Salmonella* were performed as previously described [1]. Briefly, the same mass of yolk sac fluid from each egg was aseptically placed into a sterile plastic bag containing 100 mL of buffered peptone water (BPW). The BPW mixture was then incubated at 37 °C for 24 h for pre-enrichment, followed by selective enrichment with selenite–cystine (SC) and Rappaport–Vassiliadis (RV) broth. Cultures were streaked onto Xylose Lysine Terpineol 4 agar (XLT4; Hope Bio-Technology, China) and incubated overnight at 37 °C.

The isolates were reconfirmed as *Salmonella* spp. by polymerase chain reaction (PCR) targeting the stn gene, using a method previously described [21]. The program for PCR amplification was 25 cycles of 94 °C 1 min; 55 °C 1 min, and 72 °C 1 min. Sequences of the synthesized primers were Stn-1 (5′-CTTTGGTCGTAAAATAAGGCG-3′) and Stn-2 (5′-TGCCCAAAGCAGAGAGATTC-3′), and the amplified fragments of 260 bp were detected by agarose gel electrophoresis.

### 2.2. Microbiological Analysis

The purified *Salmonella* isolates were identified by a commercial *Salmonella* biochemical assay (Haibo Technology Corporation, Qingdao, China). Serotyping was performed with a *Salmonella* diagnostic serum kit by agglutination (Tianjin Biochip Corporation, Tianjin, China) according to the manufacturer’s instructions. The antigenic formulas were used in the Kauffman–White classification to determine the specific serotype [22].

### 2.3. DNA Extraction and Whole Genome Sequencing

Genomic DNA was extracted using a TIANamp bacteria DNA kit (Tiangen Biotech, Beijing, China) according to the protocol of the manufacturer. DNA concentration and purity were determined via a Qubit fluorometer and Nanodrop 2000 spectrophotometer (Thermo Fisher Scientific, Carlsbad, CA, USA). DNA integrity was assessed by electrophoresis using a 0.5% agarose gel [23]. The genomes of 14 isolates were sequenced using an Illumina HiSeq 4000 System (Illumina, San Diego, CA, USA) at the Beijing Genomics Institute (Shenzhen, China) [24]. Genomic DNA was randomly sheared using a Bioruptor ultrasonicator (Diagenode, Denville, NJ, USA) to construct three read libraries. Raw reads of low quality from paired-end sequencing (those with consecutive bases covered by fewer than five reads) were discarded. The sequenced reads were assembled using SOAPdenovov1.05 software (Beijing Genomics Institute, Beijing, China) [25,26].

### 2.4. Molecular Typing and Phylogenetic Tree

To further genotype Salmonella isolates, the whole genome of strains was uploaded to the PubMLST database (https://pubmlst.org/ (accessed on 8 October 2021)) for multilocus sequence typing (MLST) and cgMLST analysis [27]. Salmonella housekeeping gene sequences were downloaded from PubMLST, including *aroC*, *dnaN*, *hemD*, *hisD*, *purE*, *sucA*, and *thrA*. A local BLAST database was generated from the seven housekeeping and 16S rRNA gene sequences using BLAST v2.12.0, and the most similar segments were clipped from the alignments with the whole genomes of the *Salmonella* strains. These strains included *S.* Thompson isolates, other common *Salmonella* of the same or different serotypes found in China, and those from the NCBI BioSample database (https://www.ncbi.nlm.nih.gov/ (accessed on 17 February 2021)). After aligning the two ends of the fragment sequence of each gene, all the target genes were spliced into one sequence using Python (Python version 3.9). Each concatenated sequence of the seven housekeeping genes and 16S rRNA were aligned by ClustalW (https://www.ebi.ac.uk/Tools/msa/clustalo/ (accessed on 22 October 2021)) using the order 16SrRNA-*thrA*-*dnaN*-*aroC*-*sucA*-*hisD*-*hemD*-*purE*. The phylogenetic trees of the 29 *Salmonella* strains were built based on the concatenated sequences using the neighbor-joining (NJ) method by Mega-X [28], the bootstrap test using 500 replicates.

### 2.5. Resistance Genes and Virulence Factors

The whole genomes of all *S.* Thompson isolates were uploaded to the databases ResFinder 4.1 of the Center for Genomic Epidemiology (https://cge.cbs.dtu.dk/services/ResFinder/ (accessed on 2 October 2021)) [27] and the Virulence Factor Database (VFBD; http://www.mgc.ac.cn/VFs/ (accessed on 2 October 2021)) for predictions of resistance and virulence genes [29].

### 2.6. Antimicrobial Susceptibility Testing

Antimicrobial susceptibility tests of 18 antibiotics were performed using the classical agar dilution method on Mueller-Hinton agar (Hope Bio-Technology, Qingdao, China) based on the Clinical and Laboratory Standards Institute (CLSI) guidelines (M100S-S26) [30]. *Escherichia coli* ATCC 25922 was used as the quality control strain. *E. coli* ATCC 25922 and *S.* Thompson isolates were grown in MH broth or agar at 37 °C. The antibiotics used in testing and their concentration ranges (μg/mL) are presented in Table 1.

### 2.7. Experimental Inoculation

The most resistant isolate, *S.* Thompson SDD8E033, and quality control strain *S.* Enteritidis (ATCC13076) were used to inoculate the 1-day-old specific-pathogen-free (SPF) chicks. The 24 chicks ready for inoculation were divided equally into four groups and placed into four separate sterile boxes. Groups STa, STb, SE, and placebo chicks were inoculated with 0.2 mL of 10^9^ CFU/mL *S.* Thompson SDD8E033, 10^8^ CFU/ mL *S.* Thompson SDD8E033, 10^8^ CFU/ mL *S.* Enteritidis ATCC13076, and normal saline, respectively, through the yolk sac. The umbilicus and surrounding area of each chick were disinfected on a sterile ultra-clean table. Near the umbilicus of the chick, 0.2 mL of inoculum or saline was injected into the yolk sac with a disposable syringe, with the needle inserted approximately 2–3 cm, slightly toward the cloaca [31].

The chicks were reared in wire cages at an ambient temperature of 30 °C and allowed free access to water and pathogen-free feed. The staff at the animal facility complied with all necessary biosecurity measures. The chicks were monitored at regular intervals, and any clinical manifestations were recorded. The food consumption of each group and the body weight of each chick were recorded daily. Feces were collected, and *Salmonella* loading was calculated daily.

### 2.8. Sampling and Necropsy

Feces of the same quality were collected from each group at the same time every day for *Salmonella* identification and the proportion of *Salmonella* in feces. The presence of *Salmonella* was detected by XLT4 agar and PCR, followed by serotyping as described in the sections “Strains” and “Microbiological analysis”. The proportion of *Salmonella* in feces was determined by dividing the total *Salmonella* in the feces grown on XLT4 agar by the total of all bacteria in the feces grown on Luria-Bertani (LB) agar containing 5% serum. The flat colony counting method was used to count bacteria after a decimal dilution. Once the chicks died after infection, necropsies were performed immediately. The rest of the surviving chicks were euthanized and underwent necropsy at 72 h post-infection. After the necropsy, thymus, heart, liver, spleen, intestine, yolk sac, and bursa phalloides were homogenized separately in 2 mL of a sterile 0.9% NaCl solution at equal concentrations. Each organ was tested for *Salmonella* using the same methods as the feces samples. The liver, intestine, spleen, and thymus tissues were fixed with a 4% paraformaldehyde solution, and histopathological changes were analyzed by hematoxylin and eosin (H&E) staining.

### 2.9. Statistical Analysis

Statistical analysis was performed using GraphPad Prism 8.2.1 software and R software (R version 4.0.5). The survival curve was analyzed using the log-rank (Mantel–Cox) test (*p* < 0.05). With the premise that the variance test satisfies the normal distribution, the data of weight change over time were analyzed using MANOVA for repeated measures, and the treatment factor *p*-value (*p* < 0.05) reflects the statistical significance between different treatment groups.

### 2.10. Ethics Statement

This animal experiment and management were performed according to the Animal Care and Use Committee of the Feed Research Institute of the Chinese Academy of Agricultural Sciences and approved by the Laboratory Animal Ethical Committee and its Inspection of the Feed Research Institute of CAAS (AEC-CAAS-20090609). One-day-old SPF chickens weighing 30–40 g from Boehringer Ingelheim (Beijing, China) were acclimated for 3 h before use.

## 3. Results

### 3.1. Biochemical Characteristics and Serotypes

The colony morphology of 14 *S.* Thompson strains on XLT4 agar was a medium-sized transparent round colony with a black center. The bacterial body was red, flat-ended, and rod-shaped under an optical microscope. Biochemical experiments showed that these Salmonella isolates fermented lysine, ornithine, glucose, mannitol, galactitol, xylose, sorbitol, and citrate; produced hydrogen sulfide; and were motile. The *Salmonella* serotype was verified by the antigen agglutination test according to somatic (O) and flagellar (H: H1 and H2) antigens. The antigen formula of all isolates was 7:k:5, which was identified as *Salmonella* enterica subsp. enterica serovar Thompson with reference to the Kauffman–White scheme.

### 3.2. Phylogenetic Tree, MLST and cgMLST

A phylogenetic tree of 29 *Salmonella* strains based on the 16S rRNA and seven housekeeping genes was constructed to evaluate the relationship between *S.* Thompson in this study and other *Salmonella* of the same or different serotypes. Except for NZ_CP061118.1 and NZ_LSHA000000000, all other strains clustered on the phylogenetic tree. The phylogenetic tree showed that the genetic distance within 14 of the *S.* Thompson isolates was very close, and eight had a genetic distance of zero. The evolutionary relationship between the 14 *S.* Thompson isolates and the other 15 *Salmonella* strains was relatively long. Clinical isolates of *S.* Thompson were more closely related to *S.* Thompson in the database than to different serotypes of *Salmonella*.

The MLST and cgMLST of the 29 *Salmonella* strains indicated that the 14 strains of *S.* Thompson belonged to sequence type (ST) ST26 and cgST-12774, respectively. The MLST of all the *S.* Thompson strains was ST26, but the cgMLST was not consistent (Figure 2). The *S.* Thompson strain gene sequence in this study was consistent with CP041171.1 (Shanghai, China, 2011) and CP028729.1 (Hefei, China, 2014) and was different from the foreign *S.* Thompson strains (Figure 2).

### 3.3. Antimicrobial Resistance Genes and Virulence Genes

Comparison with the ResFinder database revealed that the 14 *S.* Thompson isolates only carried *aac(6′)-Iaa* and *mcr-9* resistance-related genes. *Mcr-9*, a polymyxin resistance-related gene, was only present in isolates and not in the reference strains (Figure 2). We also found that 14 *S.* Thompson isolates had the same number and types of virulence genes, including *Salmonella* pathogenicity island (SPI)-1 (*mogA*, *sitC*, *hilA*, *invH*, *sicA*, *iacP*, *sipA*, *prgK*, *avrA*, and *prgH*), SPI-2 (*sscA*, *sseE*, *sseC*, *ssaQ*, *ttrC*, and *sifA*), SPI-3 (*mgtC* and *misL*), SPI-4 (*bcfA* and *orf319*), SPI-5 (*araB*, *sopB*, and *pipC*), virulence regulators (*phoP* and *rpoS*), type I fimbriae gene (*fimA*), flagellar assembly genes (*bcfC*, *flhD*, *fliA*, *fliC*, *fljB*, *flgK*, and *lpfC*), and other virulence genes (*iroN*, *pagC*, and *cigR*).

### 3.4. Antimicrobial Susceptibility

The minimum inhibitory concentration (MIC) of the 14 strains of *S.* Thompson to 18 antibiotics indicated resistance against streptomycin (100%), ampicillin (35.7%), and doxycycline (14.3%), with intermediate resistance against apramycin (100%), florfenicol (50%), and enrofloxacin (21.4%) (Figure 3). Of these, there were two isolates (*S.* Thompson SDD8E033 and *S.* Thompson SDD8E038) resistant to three classes of antibiotics and thus defined as multidrug-resistant bacteria.

### 3.5. Clinical Monitoring

No clinical signs were found in the chicks receiving the placebo. The clinical signs displayed by the inoculated chicks differed according to their groups. All the chicks in the STa group died rapidly within one day, stood unsteadily, had difficulty breathing when they fell to the ground, experienced opisthotonos, and passed white loose stools. The clinical symptoms of the STb group and the SE group were similar. The chicks were depressed, unsteady, and discharged gray feces. However, the chicks of the STb group died faster (*p* = 0.0004) than the SE group (Figure 4A). Compared with the control group, the mean body weight of the chicks in the STb and SE groups increased slowly after inoculation (*p* < 0.05, P_1_ = 0.0313, P_2_ = 0.0037); the STa group that died within one day was not considered (Figure 4B). The feed intake of the three groups inoculated with bacteria showed a downward trend, but the SE group returned to normal on day 3 (Figure 4C). Except for the placebo group, the other three groups showed an increase in the *Salmonella* proportion in feces (Figure 4D).

### 3.6. Necropsy and Pathological Changes

The autopsy results demonstrated that the yolk sacs of dead chicks inoculated with clinical S. Thompson were large and not absorbed by the organism, the vessels on the peritoneum were hyperemia, the contents appeared liquefactive necrosis, and the viscosity of the yolk fluid was reduced (Figure 5B). The intestinal wall was thinning, and the liver was slightly enlarged and had a yellow surface with red streaks. The spleen and gallbladder were moderately enlarged, and the bile was thick black and green. Some chicks in the STb group developed fibrinous pericarditis.

The main lesion sites (liver and intestine) and important immune organs (spleen and thymus) were selected for pathological sections to observe pathological changes. Based on the pathological section, the pathological changes of various organs caused by *S.* Thompson and *S.* Enteritidis (quality control strains) were different. After being inoculated with *S*. Thompson, the organs of the chicks showed extensive steatosis and hepatic vascular congestion of the liver, necrosis of the intestinal mucosa, necrosis of the spleen capsule, and bacterial colonies in the interstitium. Thymus lesions were mainly caused by congestion. The intestinal pathology caused by *S.* Enteritidis was similar to that caused by *S.* Thompson, but the other organ pathologies were different. In *S.* Enteritidis-treated chicks, the liver was mainly infiltrated by lymphocytes, the spleen was congested, and thymocytes were necrotic and degenerated. In each group of inoculations, the inoculated strain was isolated from the diseased organs (Figure 5A).

## 4. Discussion

In this study, the isolates from dead poultry embryos at the incubation end of the poultry farm provided us with an opportunity to systematically study pathogenic avian *S*. Thompson.

According to our investigation, the mortality rate of this batch of eggs in the broiler breeding farm was 25% during hatching, and only one serotype of *Salmonella* was isolated from the yolk sac; therefore, it is likely that *S*. Thompson was the cause of the significant decrease in the successful hatching rate. In terms of biochemical characteristics and antigenic structure, the colonies of the 14 *S*. Thompson strains were similar to *S.* Thompson isolated from dead chicks on a chicken farm in Ancient Hohhot, Inner Mongolia, reported by Chen Dewei [32].

Evolutionary analysis was performed after the strain identification. Due to the mismatch between the evolutionary rate of genes and that of the species, analysis of the evolutionary relationship of 16S rRNA genes alone is limited [33]. Therefore, in this study, multilocus sequence analysis was performed on the 29 *Salmonella* strains using 16S rRNA and seven housekeeping genes. Several researchers have demonstrated that the evolutionary analysis of a housekeeping gene was equal to or even superior to 16S rRNA analysis alone [34,35]. In the phylogenetic tree, the isolates showed close kinship, and eight isolates can be clustered to the same clone. Therefore, the 14 strains of clinical *S.* Thompson may have evolved from a clone. Interestingly, the overall confidence level of all branches of the phylogenetic tree was high, but both the isolated *S.* Thompson and the reference *Salmonella* from the NCBI database showed that *Salmonella* strains of the same serotype were instead more distantly related than that of different serotypes. Therefore, whether the building method based on housekeeping genes can better classify *Salmonella* strains needs to be further investigated.

Zhujun et al. [36] performed core genome single-nucleotide polymorphism (SNP) typing of 125 representative N. meningitidis strains and compared the results to the MLST and cgMLST typing that have been published on the pubMLST website. Their results confirmed that both the core genome SNP typing and cgMLST were significantly better than MLST. In our research, MLST of all *S*. Thompson isolates (including the reference strain) was ST26, while cgMLST results were different, similarly indicating that cgMLST was more accurate. The cgMLST of 14 strains of *S.* Thompson isolates was cgST-12774, indicating that these isolates may have evolved from the same clone. Most notably, the cgMLST of the current *S.* Thompson isolate was cgST-12774, the same as the two human-derived *S.* Thompson reference strains in China. They were CP028729.1 *S.* Thompson str. HFCDC-SM-846 and CP041171.1 *S.* Thompson str.SH11G0791, which were from Hefei (2014) and Shanghai (2011), respectively, and are the only two cases of human infection with S. Thompson reported in China in the NCBI database. It can be speculated that compared with MLST, cgMLST is more sensitive to small variations among strains, and its resolution can even reach the clonal level [11]. Therefore, the same cgMLST typing suggests that *S.* Thompson, which is pathogenic to humans and poultry in China, belongs to the same molecular type and may infect each other through the food production chain, requiring further monitoring.

For the study of antimicrobial resistance, both genotyping and phenotyping were carried out. Resistance genes can predict the antimicrobial resistance phenotype of bacteria, and a correlation has been demonstrated between WGS-based bacterial resistance prediction results and the antimicrobial resistance phenotype of bacteria in multiple studies [37,38]. In this study, two resistant genes, *aac (6′)-Iaa* and *mcr-9,* were predicted by uploading the WGS sequence data of 14 *S.* Thompson to the antimicrobial resistance gene database. The other reference strains in the phylogenetic tree had resistance genes but not *mcr-9* gene. Genes *aac(6′)-Iaa* cause the acetylation of aminoglycoside antibiotics at the 6′ amino group, leading to broad-spectrum aminoglycoside resistance [39]. The presence of *aac(6′)-Iaa* echoed the antimicrobial resistance phenotype of *S.* Thompson isolates by streptomycin (100%) in susceptibility testing. Another *mcr-9* gene located on the *IncHI2* plasmid, one of the polymyxin resistance genes, was reported by Carroll [40] in 2019 in a multidrug-resistant strain of *S.* Typhimurium. Carroll found that *mcr*-*9* did not cause the phenotypic resistance of *S.* Typhimurium to colistin, which was consistent with our results. However, when it was cloned into colistin-sensitive *Escherichia coli*, *mcr-9* showed colistin resistance. Another study showed that *mcr*-*9* had little effect on polymyxin sensitivity when *E*. *coli* was in an uninduced state, and so the clinical relevance of *mcr-9* depended on polymyxin induction [41]. However, polymyxin is an antibiotic of last resort for the treatment of severe infections caused by multidrug-resistant and extensively drug-resistant bacteria [42]. The *mcr-9* gene found in isolates was first reported in *S.* Thompson. Notably, *S.* Thompson has a broad host range and can easily proliferate in various foods, and is therefore likely to spread along the food production chain with other *Enterobacteriaceae* through horizontal gene transfer, posing a threat to public health.

Virulence of *Salmonella* is related to SPIs, virulence plasmids, pili, flagella, enterotoxin, and other factors. SPI is involved in the invasion of intestinal epithelial cells, and the important SPIs are SPI-1 through SPI-5 [43], encoding the type III secretion system, the killing effect of bacterial escape from macrophages, the survival of *Salmonella* in host macrophages, a type I secretion system, intestinal mucosal fluid secretion, and an inflammatory response, respectively [44,45]. Virulence plasmids play an important role in the stage of systemic *Salmonella* infection of the host, and the *spv* gene is a commonly found virulence gene. Type I (*fim*) pili help the thalli adhere to the surface of animal cells, promoting their colonization. Flagella is the main motor organ of *Salmonella*. Flagellar proteins, such as *fliC*, have a role in the formation of biofilms and the induction of the immune and inflammatory response of the host. Regulatory genes such as *phoP*-*phoQ* regulate virulence genes encoded by different virulence islands. The virulence of bacteria is greatly reduced if these regulatory genes are knocked out. The 14 *S.* Thompson isolates in the present study contained the same virulence genes. Virulence genes related to SPI1-5, type I pili, flagella, and regulatory genes were found in the *S.* Thompson isolates. The presence of these genes, *fimA*, *invH*, *phoP*, and *fliC*, may be one of the reasons for the strong pathogenicity of *S.* Thompson isolates.

Compared to *Salmonella* Pullorum and *Salmonella* Gallinarum, which are highly pathogenic to all growth stages of poultry, NTS systemic infections are generally more pathogenic to chicks [2,46,47]. In 1989, the chicks of four poultry farms in Beijing and Tianjin broke out with salmonellosis of *S.* Thompson, with the disease presenting at 2 days of age and a large number of deaths occurring at 5 days of age [32]. The yolk sac enters the abdominal cavity towards the end of the incubation period, with the remaining yolk in newly hatched chicks about 20% of the chick’s body weight [48]. The wound in the open umbilical area permits entry to the yolk sac, putting chicks at risk of various bacterial diseases. Therefore, our findings on the pathogenicity of *S.* Thompson should be implemented on newborn chicks. Stephens [31], in exploring the relationship between *Salmonella* growth and pathogenicity in the yolk sac of chickens, injected 1 mL of three species of *Salmonella* inoculum into the yolk sac of newly hatched chicks with a 2.54 cm subcutaneous syringe through the navel at a dose of 3.5 × 10^6^ CFU. Ultimately, the growth patterns of the three *Salmonella* species were nearly identical, but *Salmonella* Heidelberg was found more frequently in the liver and caused much higher mortality than *Salmonella* Anatum or *Salmonella* Infantis. In this study, the inoculation method was yolk sac inoculation, referred to Stephens [31], which better reflects the hazard of umbilical contamination in newborn chicks than oral and intraperitoneal inoculation. One-day-old SPF chicks with large yolk sac volumes were selected for bacterial inoculation experiments to explore the pathogenicity of *S.* Thompson. Accordingly, inoculation with approximately 0.2 mL of *Salmonella* at a concentration of 10^8^ CFU/mL ensures a minimum inoculation of more than 2 million cells while avoiding additional effects on the chicks due to the high injection volume. The results showed that the greater the number of challenge bacteria, the higher the fatality rate. Compared with *S.* Enteritidis at the same challenge dose, *S.* Thompson was the dominant serotype for chicks with a higher fatality rate and more diseased organs.

## 5. Conclusions

In this study, evolutionary analysis, molecular typing, antimicrobial resistance, and virulence genes based on WGS were performed on 14 clinical *S.* Thompson strains, as well as antimicrobial sensitivity tests for 18 antibiotics and bacterial inoculation experiments on one-day-old chicks. Overall, we came to three conclusions: (1) the significant decrease in embryo hatching rate in this farm was caused by *S.* Thompson, which was indeed highly pathogenic to newborn chicks; (2) fourteen clinical *S.* Thompson strains probably originated from the same clone, and the colistin (polymyxin B) resistance gene *mcr-9* was reported for the first time in *S.* Thompson; (3) the identical cgST-12774 typing of *S.* Thompson of human and avian origin in China suggests that we need to pay more attention to food hygiene and safety, as well as establish a more accurate typing and monitoring system for *Salmonella*.

## Figures and Tables

**Figure 1 vetsci-09-00349-f001:**
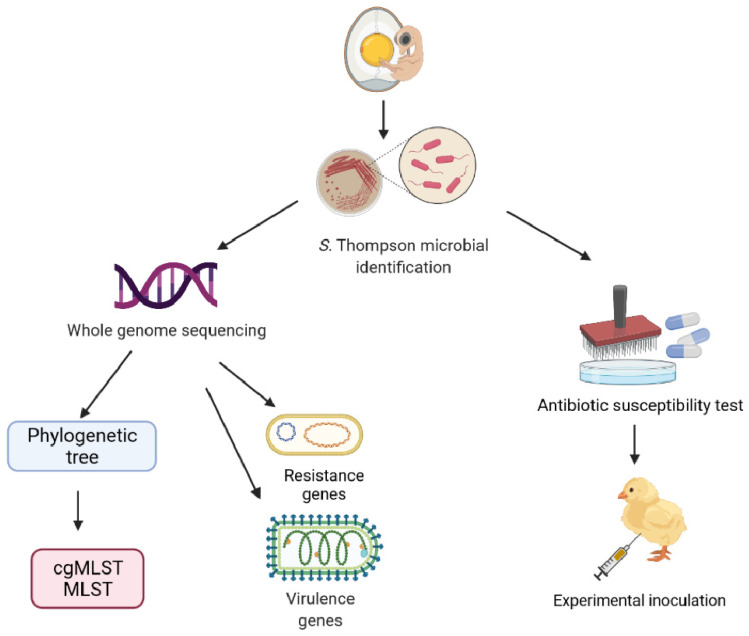
The technical route of the poultry source *S.* Thompson system research.

**Figure 2 vetsci-09-00349-f002:**
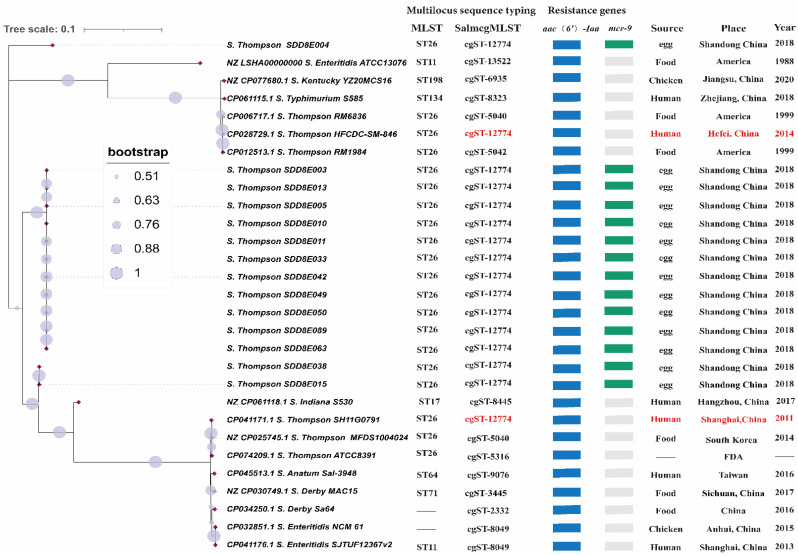
Phylogenetic tree of 14 *S.* Thompson isolates and 15 strains of different serovars from NCBI based on 16S rRNA and 7 house-keeping genes. MLST, cgMLST, and resistance genes (aac (6′)-Iaa and mcr-9) of 29 Salmonella strains. The phylogenetic tree percentages were obtained using the bootstrap test using 500 replicates.

**Figure 3 vetsci-09-00349-f003:**
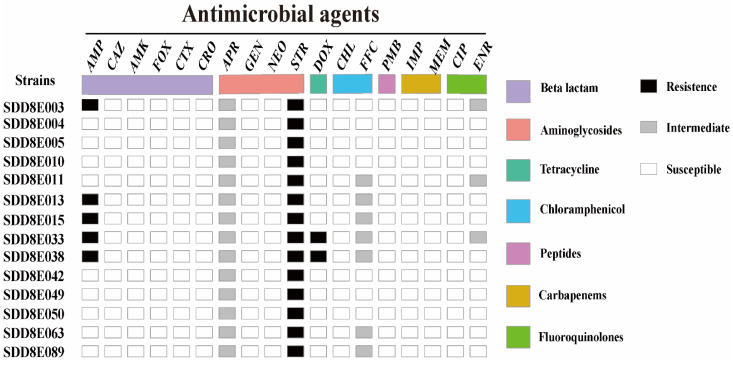
Susceptibility of all Salmonella isolates to 18 antibiotics.

**Figure 4 vetsci-09-00349-f004:**
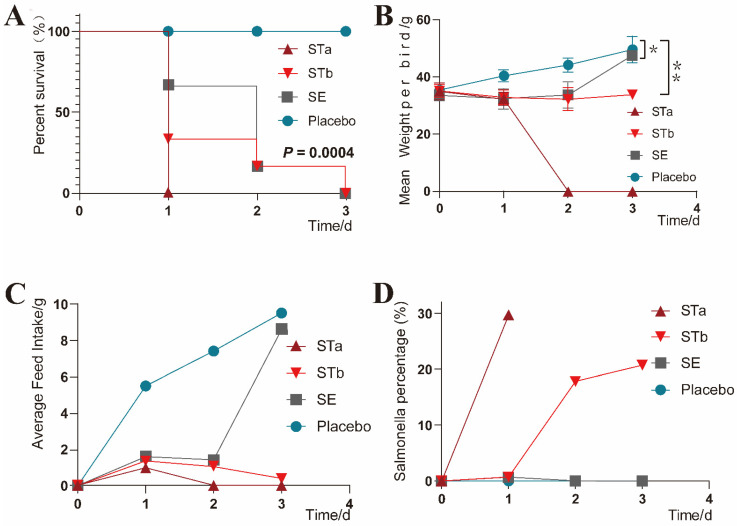
Quantitative analysis of physiological indices before and after experimental inoculation. (**A**) Survival curve plots of four sets of SPF chick experiments during 3 days (log-rank test, *p* = 0.0004). (**B**) Mean daily weight curves of each bird of each group during 3 days (Mean ± SD, MANOVA of repeated measures, *p* < 0.05, *p*_1_ = 0.0313, *p*_2_ = 0.0037). * *p* < 0.05 ** *p* < 0.01 (**C**) The curves of average daily feed intake of each group. (**D**) Daily changes in the proportion of *Salmonella* in feces.

**Figure 5 vetsci-09-00349-f005:**
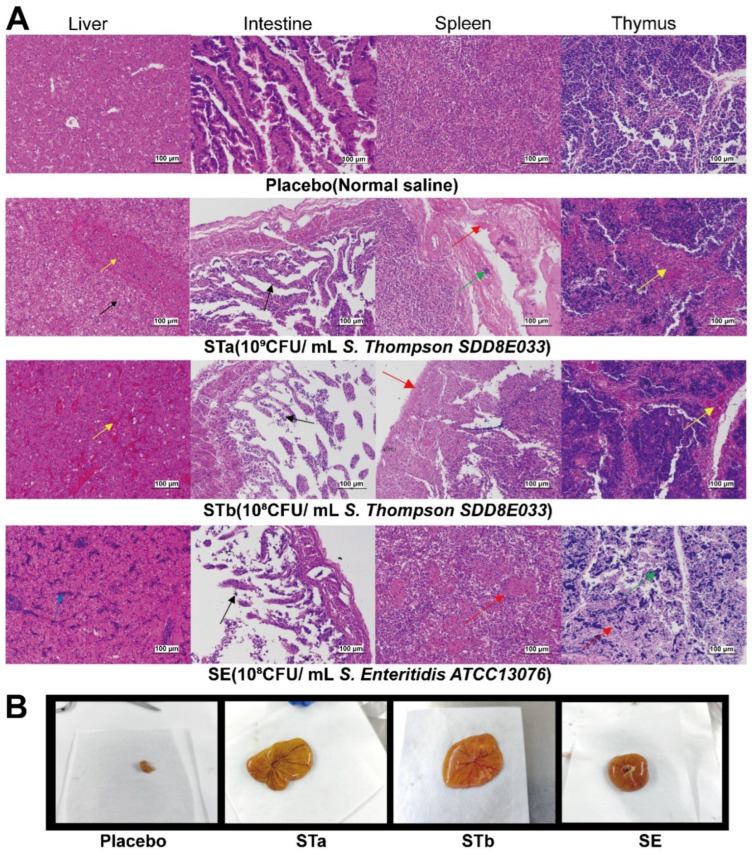
Pathological changes of the main organs and autopsy of yolk sacs following experimental inoculation. (**A**) Yellow arrow: stasis, black arrow: vesicular degeneration of hepatocytes or sloughing of intestinal epithelial cells, blue arrow: infiltration of inflammatory cells, red arrow: cytoplasmic shrinkage or nuclear pyknosis, green arrow: calcified nodules. (**B**) The yolk sacs of each group were placed on a weighing paper with a side length of 8.9 cm. Compared to the placebo group, the yolk sacs in the STa, STb, and SE groups showed incomplete absorption, vascular congestion on the sac membrane, liquefied necrosis of the contents, and reduced viscosity of the yolk fluid.

**Table 1 vetsci-09-00349-t001:** Antimicrobials and the range of concentrations tested.

Antibiotics		Abbreviation	Concentration Range (μg/mL)
β-lactam	Ampicillin	AMP	0.5~256
	Ceftazidine	CAZ	0.25~128
	Ceftriaxone	CRO	0.06~32
	Cefotaxime	CTX	0.06~32
	Cefoxitin	(FOX	0.5~256
Aminoglycosides	Amikacin	AMK	1~512
	Apramycin	APR	0.25~128
	Gentamicin	GEN	0.25~128
	Neomycin	NEO	1~512
	Streptomycin	STR	1~512
Tetracycline	Doxycycline	DOX	0.25~128
Chloramphenicol	Chloramphenicol	CHL	1~512
	Florfenicol	FFC	0.25~128
Polypeptide	Polymyxin B	PMB	0.06~32
Carbapenems	Imipenem	IMP	0.06~32
	Meropenem	MEM	0.06~32
Fluoroquinolones	Ciprofloxacin	CIP	0.015~8
	Enrofloxacin	ENR	0.015~8

## Data Availability

The datasets of 14 *S.* Thompson generated for this study can be found in the NCBI BioProject with the accession numbers PRJNA806689, PRJNA806731, PRJNA806732, PRJNA806733, PRJNA806735, PRJNA806736, PRJNA806738, PRJNA806740, PRJNA806741, PRJNA806742, PRJNA806743, PRJNA806744, PRJNA806745, and PRJNA806746.

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
