# Peer review of "Resistance and Pathogenicity of Salmonella Thompson Isolated from Incubation End of a Poultry Farm"

_vetsci, 2022, doi:10.3390/vetsci9070349_

Round 1
Reviewer 1 Report
Dear Authors,
Thank you for submitting this interesting study that investigates the evolution and pathogenicity of Salmonella in chickens. Your study is useful and well developed. At current there are a few gaps in the methods regarding animal husbandry and environment. Please also consider your test choice and it’s assumptions (ie normality). I have provided a pdf version of your paper with specific comments for consideration.

Author Response
Thank you for your hard and meticulous work, and we have carefully revised your suggestions and issues. However, in order to show the revisions more clearly, we are replying to you on the original PDF, so that you can easily check the revisions in the new document against the original location. If there is any inconvenience, you can point out it as much as you like, we can re-aggregate the issues to reply.

Reviewer 2 Report
Line 31-32: I checked these references and they do not mention the early stages of breeding, please review these. Also what do you mean by the early stages of breeding? Incubation and hatching or with reference to breeding birds?
Line 54-Please add references for the embryonic mortality caused by S. Thompson. Also describe the lesions that were associated with infection.
Line 146-147: why did you select these two concentrations for inoculation?
Line 166: which organs were selected for the homogenate? Were all of the organs pooled per bird or homogenated separately?
Line 169: at what age did the birds generally die after infection? If the yolk sac was present, why wasn’t it one of the organs that was examined microscopically?
Figure 2: this figure has a lot of information, please condense the information and add some of the additional information to supplementary figures, for example, source, place, year can be added to a supplementary table
Line 258: did you mean watery and turbid? Were there any other signs of infection such as hyperemia, discoloration, serositis?
Line 258-259: how much enlarged for the liver and spleen? Mild moderate or severe?
Line 277, the figure shows changes that I would consider autolytic and as due to processing artifact. These images do not seem representative of true pathology.
Figures: Please take higher magnification images that can better show the lesions that are being described. What is being shown with the yolk sacs? Please describe in the caption.
Discussion
Line 389-393 I would suggest removing this statement since it is misleading. Are there any references to support this? I so please add
Author Response
Thank you for your hard and meticulous work, and we have carefully revised your suggestions and issues. Revisions and responses can be found in the zip file.

Round 2
Reviewer 2 Report
The authors have made the relevant changes in the manuscript that are satisfactory to my comments.
For the Figures with histopathology. I still feel higher magnification images are needed to show properly the changes that are being described.